# Model to Measure the Readiness of University Testing Laboratories to Fulfill ISO/IEC 17025 Requirements (A Case Study)

**Era Febriana Aqidawati [1,\*], Wahyudi Sutopo [2,\*], and Roni Zakaria [2]**

[1] Laboratory of Logistics System and Business, Department of Industrial Engineering, Universitas Sebelas Maret, Surakarta 57125, Indonesia
[2] Research Group of Industrial Engineering and Techno-economics, Department of Industrial Engineering, Universitas Sebelas Maret, Surakarta 57125, Indonesia; ronizakaria@staff.uns.ac.id
\* Correspondence: erafebriana@student.uns.ac.id (E.F.A.); wahyudisutopo@staff.uns.ac.id (W.S.)

**Abstract:** Universities are considered as a source of open innovations by producing new technology. The innovations need to be tested in licensed laboratories in order to create certified products if they are to be commercialized to the market. Many universities have established laboratories that provide testing services to society and act as a revenue-generating source. Universitas Sebelas Maret (UNS) owns an accredited center laboratory that provides testing services to external parties. In addition, the university owns other laboratories in several faculties to conduct academic activities and research and yet provide testing services, but have not been accredited. Therefore, the laboratories have the potential to be developed as part of the testing service business to support the incubation process of new technology and provide testing services. In this article, we chose UNS, one of the universities in Indonesia, to develop a framework of readiness level measurement instrument, to evaluate the readiness and to suggest improvements for laboratories to achieve accreditation. A framework of laboratory readiness measurement was developed using integration of management approach and laboratory approach. Descriptive statistics were used to create a radar chart to determine the readiness level. Based on findings and evidence analysis, we proposed improvement planning using ISO 17025 and management concept.

**Keywords:** assessment framework; innovation; ISO/IEC 17025; readiness level; university testing laboratories

## 1. Introduction

Universities have played a key role in the technological, economic, and social development of countries by producing innovations and inventions through research done by academics [1–3] to perform the role of research and development to create new technology [4,5]. In other words, universities are considered as an open innovation source as a result of the research activities [6–18]. Recently, there has been rapid progression in technology [19–21] and the industrial sector [22–24]. Lots of product innovations in these sectors resulted from universities, for instance, lithium battery [25–27], electric vehicle [28,29], and traceability technology [30,31]. The innovations would be more beneficial if they were followed by commercialization. Prior to commercialization, eligible product testing in licensed laboratories is needed to create certified products [32–34]. Due to the product innovation acceleration, the provision of laboratories has been insufficient. Thus, these factors urge universities to establish laboratories [35–37] to perform the role of testing services provider [38–42] aside from the original purpose of teaching and research laboratories [43–52].

Nowadays, it has been pretty notable for universities to accomplish revenue generation in order to maintain their sustainability [3,53]. Providing services in their laboratories to perform tests for external parties can be a means to gain revenue and support the incubation process of new technology [53]. Laboratories that provide testing services must be accredited and comply with the laboratory standard (i.e., ISO/IEC 17025) to ensure their competency in performing tests [54]. ISO/IEC 17025 is a laboratory standard that contains general requirements of laboratory competence in conducting testing and/or calibration. It includes the quality management system and technical requirements of the accreditation process [55]. There are several benefits laboratories can gain by fulfilling this standard, such as the ability to produce consistent results and more satisfying performance [56], overall improvement of laboratory business [57], the ability to prove the reliability of test results and technical competence to customers [58], and enhanced competitiveness in the market by improved quality, reliability, accuracy, and consistency of products, services, and processes [54].

Indonesia is one of the countries that strive to develop World Class Universities (WCUs) [59], one of whose goals is having adequate facilities and funding [60]. Sustainable and accountable financial support is needed to reach this goal [61]. Therefore, generating revenue from a university's internationally standardized business units is an initial step that can be taken to create world-class funding [62]. Exploiting university laboratories to provide services may be a reliable move to perform this role.

Universitas Sebelas Maret (UNS) is a university in Indonesia which is pursuing WCU status. As a key step to accelerate towards this status, the university has established some business units, one of which is a center laboratory which has been accredited based on ISO 17025. This laboratory provides testing services to external parties and has been successfully running the business, having many customers [63]. Other than the center laboratory, the university has 13 laboratories located in 3 faculties which are basically used for academic activities and yet provide testing services. However, the laboratories do not comply with testing laboratory accreditation qualifications. Besides, the university laboratories are basically research laboratories that do not have good management practices although they are expected to be product certification agencies. In addition, to support the commercialization of a university's innovation product in the market, the technology readiness level of the product has to be measured first and certified by the product certification agency [64,65]. Thus, those laboratories have potential to be developed as standardized testing laboratories. With regards to WCU status, it is important for the university to establish a testing service business within the university by integrating the center laboratory and those potential laboratories to expand revenue generation and support the commercialization of new technology so that the innovation of new technology can be introduced to the market.

Each laboratory in each faculty (i.e., Faculty of Science, Engineering, and Agriculture) has a different core competence. It means that each laboratory can provide services by offering different kinds of tests and parameters. In daily practice, the laboratories have been operating independently in terms of performing testing services. In other words, there has not been interactive relationship-building between them. However, interorganizational linkages are likely to become important when an organization strives to establish a new business ecosystem [66]. Therefore, an appropriate strategy is required to support the interorganizational relationship between laboratories in the testing service ecosystem in the university.

It is important that the university improves the quality of the testing service business to commercialize the innovation product. It means that the quality of the potential laboratories needs to be improved. Implementing an ISO/IEC 17025 quality management system is an initial step that could be done to improve the laboratories' quality [67]. To accomplish this goal, the potential laboratories must be transformed into licensed laboratories by fulfilling ISO 17025 accreditation requirements. Therefore, this study aims to develop a framework to measure the readiness level of the potential laboratories to be developed as part of the testing service business by considering ISO 17025 requirements and organizational audit factors.

There have been many researchers studying about the readiness level of a system. Sutopo et al. [68] developed a framework to measure university readiness for establishing spin-offs considering the optimal time that is used to launch a spin-off company. Lokuge et al. [69] conceptualized a formative multidimensional construct to gauge organizational readiness for digital innovations. Kobos et al. [70] developed Regulatory Readiness Level (RRL) and Market Readiness Level (MRL) frameworks towards technology development to meet desired technical and policy goals in the coming decades. There also has been much research regarding ISO 17025. Rodima et al. [71] described the benefits that a university can gain from having an ISO 17025 accredited quality system in place. Meanwhile, Hullihen et al. [72] discussed experience in establishing an ISO 17025 compliant laboratory at a university and Zapata-García et al. [73] presented the experience of implementing a quality system on ISO 17025 for the accreditation of a university testing laboratory. Moreover, organizational audit factors are used as parameters to assess strengths and weaknesses of the organization and generate strategies upon the findings [74–77].

In this article, we aimed to develop an instrument to measure the university laboratories' readiness level. Based on the readiness level evaluation, we proposed improvements for the laboratories in order to achieve accreditation. The improvements of the laboratories are expected to support the commercialization of new technology in the market.

The remainder of the paper is organized as follows: Section 2 presents the methodology employed to generate the framework. Section 3 shows the result of implemented framework and presents the descriptive statistics. Section 4 discusses the result including the findings and interpretations of it. Section 5 concludes by summing up the work.

## 2. Methods

### 2.1. Designing Laboratory Readiness Measurement Instrument

In this section, we presented the stages and approaches taken to generate a questionnaire. This questionnaire was used as an instrument to measure the laboratory readiness level in fulfilling ISO/IEC 17025 requirements. The instrument designing consisted of two stages, as follows.

#### 2.1.1. Generating Sets of Matching Criteria, Subcriteria, and Activities

At this stage, we reviewed and integrated two researches as a benchmark. Organizational audit factors, which were derived from David's research [78], consist of criteria and subcriteria used in conducting internal audits in an organization. Those factors were used to assess the laboratory's internal environment situations. Meanwhile, the research of Grochau & Caten [79] was used to determine the process elements involved in providing testing services in the laboratory. Based on this research [79], we derived the activities involved in each process element along with corresponding standard items regulated in ISO/IEC 17025. The next step was comparing the criteria and subcriteria of organizational audit factors with the processes and activities elements in conducting testing services in the laboratory. This step aimed to distribute activities in the provision of laboratory testing services to matching subcriteria. The results of this approach are shown in Table 1.

**Table 1.** Organizational audit factors and laboratory's testing service activities distribution.

| Criteria | Subcriteria | Activities | Process Element | Standard Item in ISO 17025 |
|---|---|---|---|---|
| Management (C1) | SC1-Planning | Formulate policies and objectives of the quality management system | Policies and procedures planning | 4.2 |
| | SC2-Organizing | Meet organizational requirements | Management responsibility | 4.1 |
| | SC3-Staffing | • Hire, train, evaluate, and authorize staff<br>• Describe the functions and responsibilities of personnel | Personnel management | 4.1, 5.1, 5.2 |
| | SC4-Controlling | • Record and solve complaints | Complaint | 4.8 |
| | | • Record and analyze nonconformities<br>• Plan and implement corrective actions<br>• Ensure corrective and preventive actions | Nonconformity, corrective, and preventive actions | 4.9, 4.10, 4.11, 4.12 |
| | | • Plan, implement, and record internal audits | Audits | 4.10, 4.14 |
| | | • Plan, implement, and record meetings regarding management reviews | Management responsibility | 4.10, 4.15 |
| Marketing (C2) | SC5-Service sales | • Contact the client<br>• Determine requirements<br>• Make agreements regarding methods, prices, and deadlines<br>• Formulate a contract | Review requests, tenders, and contracts | 4.4 |
| Marketing (C2) | SC6-Service planning | • Providing appropriate access to the laboratory<br>• Provide guidance for preparation, packaging, and shipping of test samples<br>• Maintain communication throughout the process<br>• Looking for feedback through customer satisfaction surveys | Customer service | 4.7 |
| Operation (C3) | SC7-Resources | • Collect, receive, identify, control, protect, and receive goods tested<br>• Implement or subcontract testing<br>• Analyze test data<br>• Record, protect, and send test reports | Testing | 5.8, 4.5, 5.10 |
| | | • Evaluating suppliers<br>• Maintain approved suppliers<br>• Develop requirements for purchasing activities<br>• Carry out purchasing activities<br>• Checking items received with the desired specifications | Purchasing | 4.6 |
| | | • Supervise, control, and record environmental conditions<br>• Adjust laboratory facilities, control access, and use of laboratory areas | Infrastructure | 5.1, 5.3 |
| | | • Study, create, validate applying and use testing methods and related procedures<br>• Estimating measurement uncertainty | Method | 5.1, 5.4 |

**Table 1.** *Cont.*

| Criteria | Subcriteria | Activities | Process Element | Standard Item in ISO 17025 |
|---|---|---|---|---|
| Operation (C3) | SC7-Resources | • Study, identify, supervise, maintain, and calibrate equipment<br>• Outlines procedures for maintenance, transfer, storage, safe use, and scheduling maintenance of measurement equipment | Equipment | 5.1, 5.5 |
| | SC8-Quality | • Develop programs and procedures for equipment calibration and standard setting<br>• Participate in a collaborative study program or testing expertise and analyzing laboratory performance | External quality control | 5.1, 5.6, 5.9 |
| Operation (C3) | SC8-Quality | • Provides intermediate checks to maintain confidence in the calibration status of the tool and reference standards<br>• Develop comparisons in the laboratory<br>• Establish quality control procedures when regular use of reference materials is certified, replicates testing or repeats testing of retained samples<br>• Analyze collected data | Internal quality control | 5.5, 5.6, 5.9 |
| Management information system (C4) | SC9-Information system management | • Publish, approve, distribute, and manage quality management system documents<br>• Identify, collect, compile, store, maintain, and tidy up documents related to quality and technical details | Information management | 4.3, 4.13, 5.4, 5.10 |

### 2.1.2. Generating Sets of Indicators

At this stage, we generated laboratory readiness indicators. We highlighted criteria, subcriteria, process elements, and standard items in the previous stage to be employed in this stage. Furthermore, we referred to the explanations of each standard item in ISO/IEC 17025. The explanations of each standard item in each process element were then summarized into sets of 117 indicators of the instrument. The result of this stage is presented in Table 2.

**Table 2.** Laboratory Readiness Measurement Instrument.

| Subcriteria | Indicator |
|---|---|
| SC1-Planning | 1 Determination of management systems<br>2 Application of management systems<br>3 Maintenance of management systems<br>4 Management policy documentation<br>5 Management system documentation<br>6 Program documentation<br>7 Documentation procedure<br>8 Documentation of work instructions<br>9 Communication of documentation to all personnel<br>10 Understanding of documentation by all personnel<br>11 Availability of documentation for all personnel<br>12 Application of documentation by all personnel<br>13 Quality manual<br>14 Quality objectives<br>15 Issuance of quality policy |

**Table 2.** *Cont.*

| Subcriteria | Indicator |
|---|---|
| SC1-Planning | 16 Laboratory management commitment<br>17 Management statement for laboratory service standards<br>18 The objectives of management systems related to quality<br>19 Requirements for all personnel to understand and implement policies and procedures<br>20 Commitment to improve the effectiveness of management systems on an ongoing basis |
| SC2-Organizing | 1 Determination of organizational structure and laboratory management<br>2 Determination of management position within the parent organization<br>3 Determination of the relationship between quality management, technical activities, and supporting services<br>4 Determination of responsibility, authority, and relations between all personnel |
| SC3-Staffing | 1 The presence of managerial and technical personnel<br>2 Division of task management and technical personnel<br>3 Adequate supervision of testing staff<br>4 Technical management<br>5 Quality manager<br>6 Appointment of deputies for core management personnel<br>7 Educational qualifications<br>8 Training qualifications<br>9 Experience qualifications<br>10 Skill qualifications<br>11 Appropriate supervisors to staff undergoing training<br>12 Formulation of educational goals<br>13 Formulation of training objectives<br>14 Formulation of skill targets<br>15 Personnel training policies and procedures<br>16 Evaluate the effectiveness of training activities<br>17 The existence of contract/nonpermanent personnel<br>18 Maintenance of applicable job descriptions<br>19 Determination of authority to take samples<br>20 Determination of authority to conduct testing<br>21 Determination of authority to issue test reports<br>22 Determination of authority to give opinions and interpretation of test results<br>23 Determination of authority to operate certain equipment<br>24 Documentation of personnel authority<br>25 Documentation of personnel competence |
| SC3-Staffing | 26 Documentation of personnel education<br>27 Professional qualification documentation<br>28 Personnel training documentation<br>29 Documentation of personnel skills<br>30 Documentation of personnel experience |
| SC4-Controlling | 1 Determination of policies and procedures for resolving complaints received from customers or other parties<br>2 Documentation regarding complaints received<br>3 Documentation related to investigations of complaints received<br>4 Documentation regarding corrective actions taken<br>5 Determination of policies and control procedures for inappropriate testing work<br>6 Increasing the effectiveness of the management system on an ongoing basis<br>7 Determination of policies and procedures for carrying out corrective actions<br>8 Determination of preventive action procedures for nonconformity and policy deviation<br>9 Establishing schedules and procedures for internal audits<br>10 Periodic internal audits<br>11 Testing activities are included in the elements that must be audited<br>12 Planning and organizing audits by quality managers<br>13 The audit is carried out by trained personnel and independent of the audited activities<br>14 Schedule and procedures for laboratory management review<br>15 Periodic implementation of laboratory management reviews |
| SC5-Service sales | 1 Determination and maintenance of customer request review procedures<br>2 Determination of tender review procedures<br>3 Determination of procedures for reviewing test contracts |
| SC6- Service planning | 1 Determination of test sampling procedures<br>2 Determination of procedures for recording test sample data<br>3 Seek feedback from customers through customer surveys |

**Table 2.** *Cont.*

| Subcriteria | Indicator |
|---|---|
| SC7-Resources | 1 Subcontract testing work<br>2 Reporting on test results<br>3 Determination of transportation procedures for goods tested<br>4 Determination of procedures for receiving goods tested<br>5 Determination of procedures for handling goods tested for deterioration, loss, or damage<br>6 Determination of procedures for protecting goods tested |
| SC7-Resources | 7 Determination of procedures for storing goods tested<br>8 Determination of procedures for identifying items tested<br>9 Recording the condition of items tested when received by laboratory staff<br>10 Consultation with customers if the goods received are not in accordance with existing specifications<br>11 Determination of policies and procedures for selecting and buying services and supplies<br>12 Procedure for purchasing reagents and disposable materials<br>13 Procedure for receiving reagents and disposable materials<br>14 Procedure for storing reagents and disposable materials<br>15 Conduct evaluation towards supplier of disposable materials, supplies, and services<br>16 Energy source is adequate<br>17 Lighting is sufficient<br>18 Adequate environment<br>19 Procedure for guaranteeing accommodation conditions and laboratory environment<br>20 Procedure for access to laboratory space<br>21 Procedure for using laboratory space<br>22 Use of appropriate methods and procedures for all tests performed<br>23 Equipment operating instructions<br>24 Instructions for handling and preparing items tested<br>25 Procedure for estimating measurement uncertainty<br>26 Equipment handling procedures<br>27 Equipment removal procedure<br>28 Equipment storage procedure<br>29 Procedure for using tools<br>30 Equipment maintenance procedures |
| SC8-Quality | 1 Calibration of equipment<br>2 Equipment calibration programs and procedures<br>3 Intermediate check<br>4 Quality control procedures<br>5 Analysis of quality control data |
| SC9-Information system management | 1 Document control procedures<br>2 Quality and technical documentation control procedures<br>3 Internal audit report<br>4 Management review report<br>5 Reports on corrective actions<br>6 Preventive action report<br>7 Procedure for the protection and backup of records stored electronically |
| 9 total subcriteria | 117 total indicators |

## 2.2. Instrument Testing

At this stage, we carried out a survey using the instrument to collect data from 13 faculty laboratories in UNS. The respondents chosen to fill out the questionnaire were the heads of each laboratory. Table 3 shows the details and categories of the surveyed laboratories. The heads of the laboratory were chosen because they knew the best about their laboratory conditions including the management and technical aspects. The respondents were asked to check off the indicators which they had fulfilled.

**Table 3.** Potential laboratories categories.

| Category | Laboratory |
|---|---|
| Engineering | Material Lab (L2) |
| | Nano Bioenergy Lab (L3) |
| | Thermodynamic Lab (L4) |
| | Structural Lab (L5) |
| | Mechanics Lab (L11) |
| | Ergonomic Lab (L12) |
| | Basic Chemistry Lab (L13) |
| Agriculture | Soil Fertility Lab (L6) |
| | Soil Conservation Lab (L7) |
| | Biotechnology Lab (L8) |
| | Plant Breeding Lab (L9) |
| | Plant Physiology Lab (L10) |
| Food technology | Food Biochemistry Lab (L1) |

## 3. Results

### 3.1. Laboratory Readiness Instrument Framework

An instrument for assessing university laboratory readiness to be developed into testing laboratories was generated. This instrument framework consists of four criteria to assess the laboratory's organizational readiness, that is, the laboratory's management, marketing, operation, and management information system. The subcriteria used in assessing the laboratory's management performance were planning, organizing, staffing, and controlling. In addition, two subcriteria were used to assess the laboratory's marketing performance (i.e., selling services and service planning). Moreover, two subcriteria were used in assessing the laboratory's operation performance (i.e., resource and quality). The corresponding processes for each subcriterion are shown in Figure 1. The framework shown in Figure 1 refers to previous work [80].

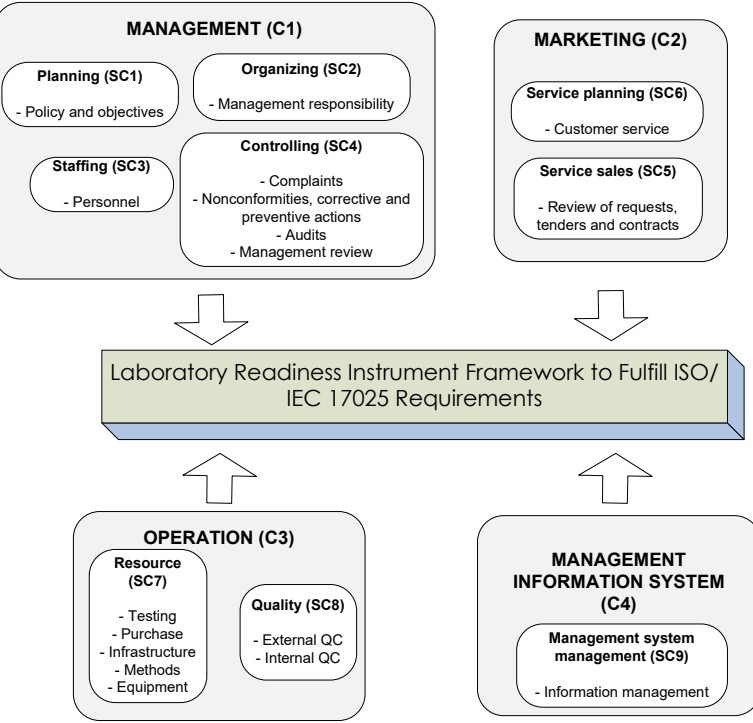

**Figure 1.** Instrument framework for assessing laboratory readiness.

## 3.2. Laboratory Readiness Level Measurement

Potential laboratories readiness level was measured by the percentage of indicators fulfilled. At this stage, a recapitulation of the number of indicators fulfilled by each respondent was conducted from a total of 117 indicators. Then, the percentage of indicators fulfilled was calculated by dividing the number of fulfilled indicators with the total indicators in the instrument. Table 4 shows the recapitulation of fulfilled indicators by each laboratory (L).

**Table 4.** Potential laboratories readiness level.

| Laboratory | Total Indicators Fulfilled | Indicators Fulfillment Percentage |
|:---:|:---:|:---:|
| L1 | 46 | 39% |
| L2 | 60 | 51% |
| L3 | 26 | 22% |
| L4 | 56 | 48% |
| L5 | 55 | 47% |
| L6 | 43 | 37% |
| L7 | 45 | 38% |
| L8 | 44 | 38% |
| L9 | 24 | 21% |
| L10 | 25 | 21% |
| L11 | 55 | 47% |
| L12 | 31 | 26% |
| L13 | 25 | 21% |

Next, we assessed the readiness level of each subcriterion by calculating the indicators fulfillment percentage of each subcriterion. The calculation was done by calculating the average percentage value for each subcriterion of the overall 13 laboratories. The result can be seen in Table 5. Figure 3 shows the percentage graph of indicator fulfillment for each subcriterion.

**Table 5.** Subcriteria readiness level.

| Criterion | Subcriterion | Indicator Fulfillment Percentage |
|:---:|:---:|:---:|
| Management | Planning | 25% |
| | Organizing | 75% |
| | Staffing | 44% |
| | Controlling | 19% |
| Marketing | Service selling | 0% |
| | Service planning | 54% |
| Operation | Resources | 45% |
| | Quality | 12% |
| Management information system | Information system management | 20% |

According to Table 5 and Figure 2, based on management aspect, the most ready subcriterion in general was organizing. Meanwhile, for the marketing aspect, in general, the laboratories were best prepared for the service planning subcriterion. While viewed from operation aspect, the laboratories were best prepared for the resources subcriterion.

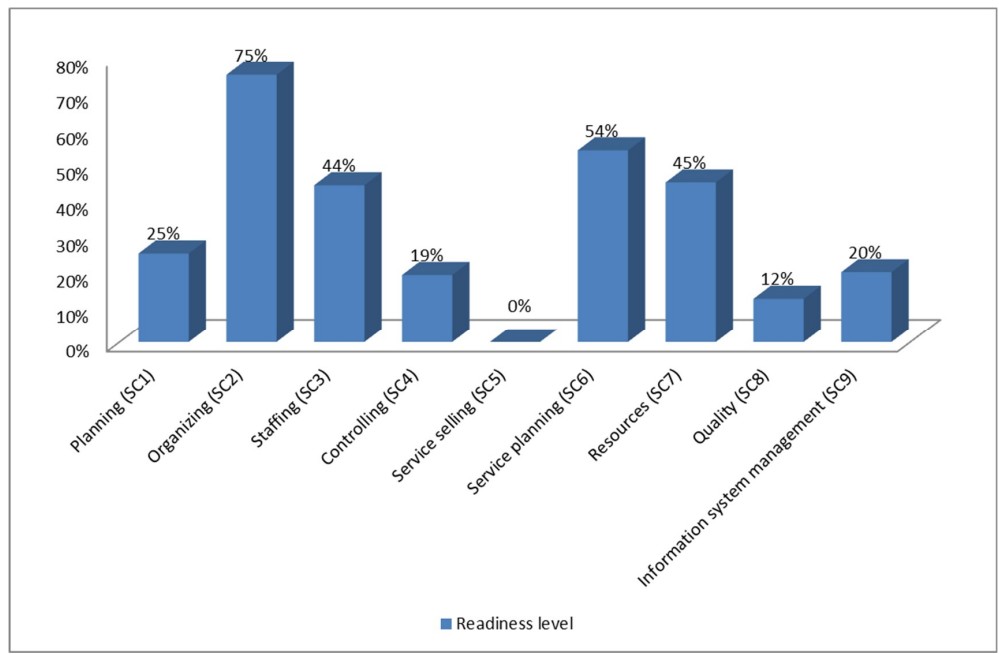

**Figure 2.** Subcriterion readiness level.

Based on the laboratories and subcriterion readiness level, we mapped the laboratories' readiness level using the radar chart to find out the subcriterion groups that tended to have high a readiness level. The readiness mapping is shown in Figure 3.

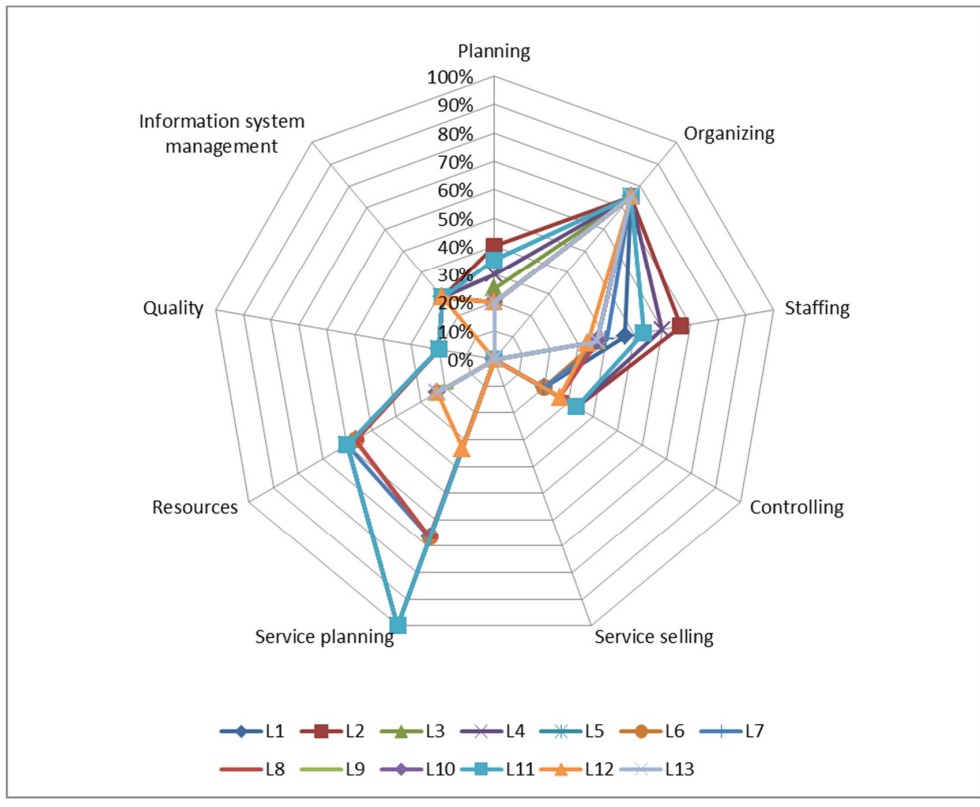

**Figure 3.** Radar chart of laboratories readiness mapping.

## 4. Discussion

A framework for assessing university laboratory readiness to be developed into testing laboratories was generated. The framework is presented in Figure 4.

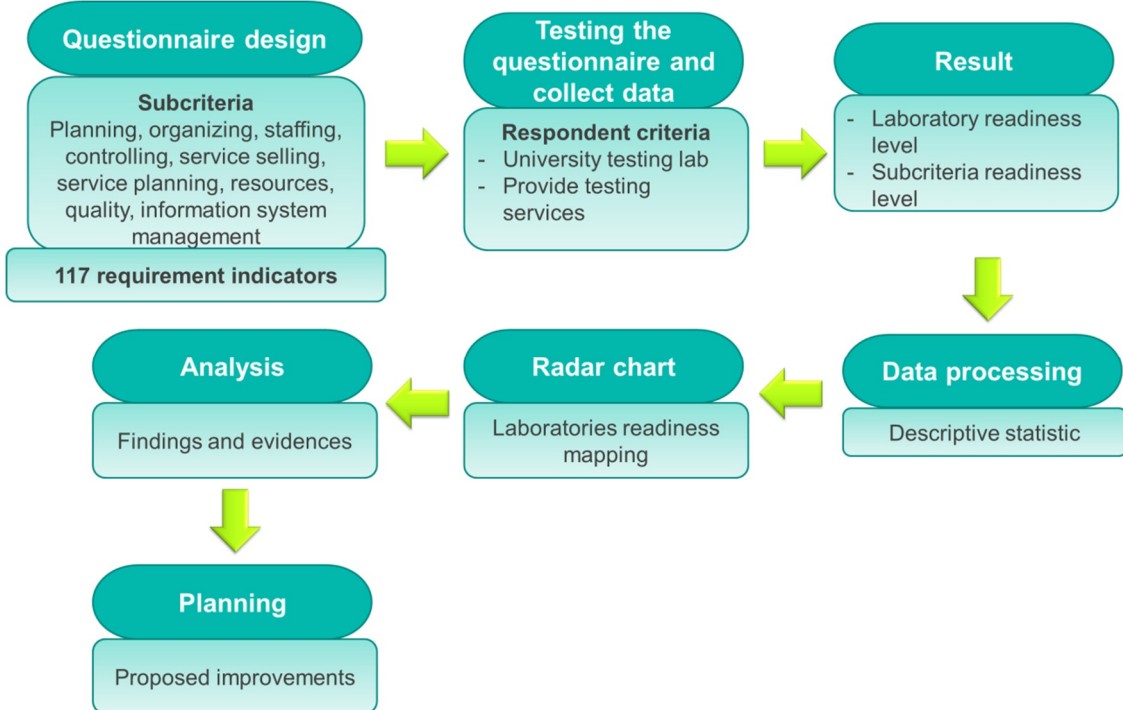

**Figure 4.** University testing laboratories readiness assessment framework.

The readiness level measurement of potential laboratories has been done. The subcriterion readiness level was also measured. Based on the results in Table 4, we analyzed the readiness level of the laboratories and we grouped the laboratories using a Pareto chart which is shown in Figure 5. There are four laboratories, namely L2, L4, L11, L5, which were identified as the most ready laboratories with the highest readiness level ranging above 40%. Based on the survey, the proportion of fulfillment in each subcriterion for these laboratories was higher compared to other laboratories. It is recommended for these laboratories to focus on improving the problems that cause the laboratory to be unable to carry out the existing requirement indicators. In addition, the laboratory is recommended to benchmark to an accredited laboratory in order to get representation of the ideal laboratory management system.

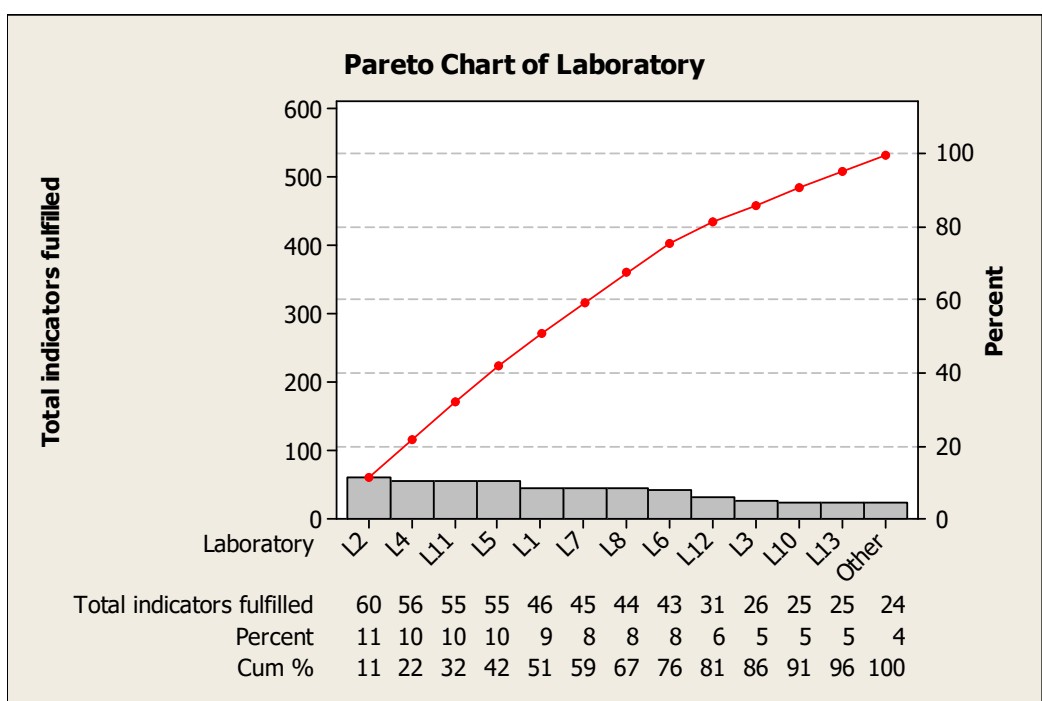

**Figure 5.** Pareto chart of laboratories readiness.

There are four laboratories, including L1, L7, L8, L6, identified with poor readiness level. Based on the survey, the proportion of fulfillment for SC1, SC3, SC4, SC7, SC8, SC9 in these laboratories was quite low. It is important for these laboratories to increase the understanding of personnel towards the existing requirement indicators, by inviting a standardization agency to socialize the requirements of ISO/IEC 17025. In addition, it is critical to make improvements to the problems that caused the laboratory to be unable to implement ISO/IEC 17025.

The final five laboratories, including L12, L3, L10, L13, L9, were identified as the least ready in fulfilling accreditation requirements as the proportion of fulfillment in each criterion was very low. In addition, these laboratories failed in fulfilling requirements in several subcriteria, especially SC4, SC5, SC6, and SC8, which had 0% in proportion of indicators fulfillment. This is because laboratory personnel do not understand and do not implement the requirements of ISO/IEC 17025. The initial strategy that can be done for these laboratories is to increase the level of readiness by improving the fulfillment of indicators in the mentioned subcriteria. This can be started by increasing the understanding of existing requirement indicators by laboratory personnel by conducting training and workshops for ISO/IEC 17025 and inviting standardization agencies to the events.

Shortly, the most ready laboratory, which is L2, was only capable of fulfilling 51% of the indicators and only three other laboratories (i.e., L4, L5, and L11) had equivalent levels. Meanwhile, the readiness level of nine other laboratories were lower. In addition, only the organizing subcriterion was assessed as the most ready, which is equal to 75%, while other subcriteria had a much lower level of readiness. This means that the university needs to do extra work and investment to encourage and strengthen these laboratories so that they can meet all the indicators that are required for the establishment of a testing laboratory service center.

In the case study, some improvements of the university laboratories in order to support services and commercialization of innovation product by using testing are needed based on the characteristics of licensed laboratories. The laboratory readiness was evaluated and there are several examples of evidence which act as weaknesses in each dimension of management factors. The findings are presented in Table 6.

**Table 6.** Existing evidence and proposed improvement.

| Dimension | Readiness Level | Evidence | Proposed Improvement |
|---|---|---|---|
| Planning (SC1) | 25% | Lack of organized management system | State quality management system policies and objectives Compose documents regarding quality guide, procedures, job instructions, and activity forms |
| | | Undocumented laboratory policies, procedures, programs, and job instructions | |
| | | Lack of quality manual | |
| | | Unstated objective and policy regarding laboratory quality | |
| Organizing (SC2) | 75% | Undivided responsibilities between technical and quality management | Recruit new personnel to carry out both responsibilities |
| Staffing (SC3) | 44% | Lack of quality manager | Recruit new qualified personnel |
| | | Lack of personnel training policy and procedure | |
| | | No relevant documentation of personnel's competency, education, qualification, training, skill, and experience | Conduct documentation compiling training for laboratories personnel |
| Controlling (SC4) | 19% | Lack of policy and procedure to resolve customer complaints | Conduct training for laboratories personnel on how to compile documents regarding laboratory quality management system and how to establish policies and procedures regarding test conduction |
| | | No documentation of customer complaints, investigation, and corrective action towards them | |
| | | No established policy and procedure to conduct corrective action | |
| | | Lack of procedure to prevent nonconformity and policy deviation | |
| | | Lack of audit planning and organizing | |
| | | Lack of schedule and procedures for laboratory management review | |
| | | Lack of periodic laboratory management review | |
| Service selling (SC5) | 0% | Lack of customer request, tender, and contract review procedure | Formulate required procedures |
| Service planning (SC6) | 54% | Low interest of laboratories to seek feedback through customer survey | Conduct customer satisfaction survey to improve service and management |
| Resources (SC7) | 45% | Lack of procedure to receive, handle, protect, and identify test items | Invest in equipment and develop testing parameters especially for prioritized product produced by the university |
| | | Lack of policy and procedure for selecting and buying services and supplies | |
| | | Lack of procedure to purchase and receive reagents and disposable materials | |
| | | No evaluation towards supplier was conducted | |
| | | Lack of procedure for guaranteeing accommodation conditions and laboratory environment | |
| | | Lack of procedure to estimate measurement uncertainty | |
| Quality (SC8) | 12% | Lack of equipment handling and calibration procedure | Formulate calibration and intermediate check procedures and make the schedule and program |
| | | No intermediate check was conducted | |
| | | Lack of quality control procedure and data analysis | |
| Information system management (SC9) | 20% | Lack of document controlling procedure, corrective and preventive action report | Formulate the required procedures and develop a management information system software to monitor the laboratory activities |
| | | Lack of record backup making and protection procedure | |

Table 6 shows the improvements that could be implemented by the laboratories. It is recommended for laboratories to create action plans by determining goals and targets for the next

period. The first stage of the plan is to improve the quality management system by focusing on problem gap improvement and fulfilling the qualifications in each dimension. During the process, monitoring and evaluation of target achievement would be needed to execute continuous improvement. The main purpose of the first stage is to actualize an appropriate management system. After fulfilling all of the qualifications, the next stage is to conduct testing by conforming to the test parameters. Related to the purpose of supporting product innovation commercialization, the laboratory has to comply with the parameters that must be tested to produce certified products. For this stage, we chose one of the technology innovations which were produced by the university, that is, the lithium battery. Based on the evaluation, it is important to improve the resource readiness by conforming the equipment and testing method to standard requirements regulated for lithium battery testing. In this case, the lithium battery must be appropriate with the three following standards:

- IEC 62660-2, Secondary lithium-ion cells for the propulsion of electric road vehicles—contains two test parameters (i.e., reliability and abuse testing)
- ISO 12405-1, Road vehicles: Electrically propelled road vehicles—contains test specification for lithium-ion battery packs and systems in high-power applications
- ISO 12405-2, Road vehicles: Electrically propelled road vehicles—contains test specification for lithium-ion battery packs and systems and high-energy application that defines tests and related requirements for battery systems

The mentioned standards above require several test parameters and machines. It is recommended to fulfill these two aspects. This could be achieved by investing in the required equipment and developing test parameters for the lithium battery.

In addition to the previously mentioned improvement strategies, it is recommended for the laboratories to form collaborative networks if the university would like to establish a testing service business center. Such a strategy would support the interorganizational relationship between laboratories in the testing service ecosystem in the university. Laboratories should cooperate with each other to share some resources and become more competitive. Resource sharing can be done by utilizing their capability in conducting different test parameters to perform tests for certain products. Some key reasons include sharing risks when entering new markets, reducing costs, and enhancing the organizational profile in selected industries or technologies.

Cooperation is a prerequisite for greater innovation, problem solving, and performance [66]. Regarding lithium battery testing, collaboration can be done by some laboratories in the university. The initial step to perform this strategy is to identify the test parameters required to conduct lithium battery testing according to mentioned standards. There are several test parameters that must be performed, including electrical measurement, measurement of cell temperature, dimension, weight, energy efficiency, and so forth. The next step is to identify test parameters that were provided by each laboratory. Then, we need to choose which laboratory will conduct which test parameter. For instance, electrical measurement is conducted by L3 and L13, dimension and weight measurement is conducted by L2, cell temperature measurement is performed by L4, and energy efficiency tests can be done by L3.

Figure 6 shows the implications provided by this study. Innovation in the technology sector can be achieved by exploiting university laboratories to conduct testing for its innovation product. To produce certified products, laboratories need to be improved using ISO/IEC 17025 and the management approach. Investment of equipment and development of testing procedures are needed to support the improvements of the quality management system, including planning, organizing, staffing, controlling, service selling, service planning, resources, and quality and information system management. Therefore, the improved laboratories would be able to provide testing of new technology and external services and the new technology commercialization could be supported. In addition, revenue generation could be supported in regard to WCU status.

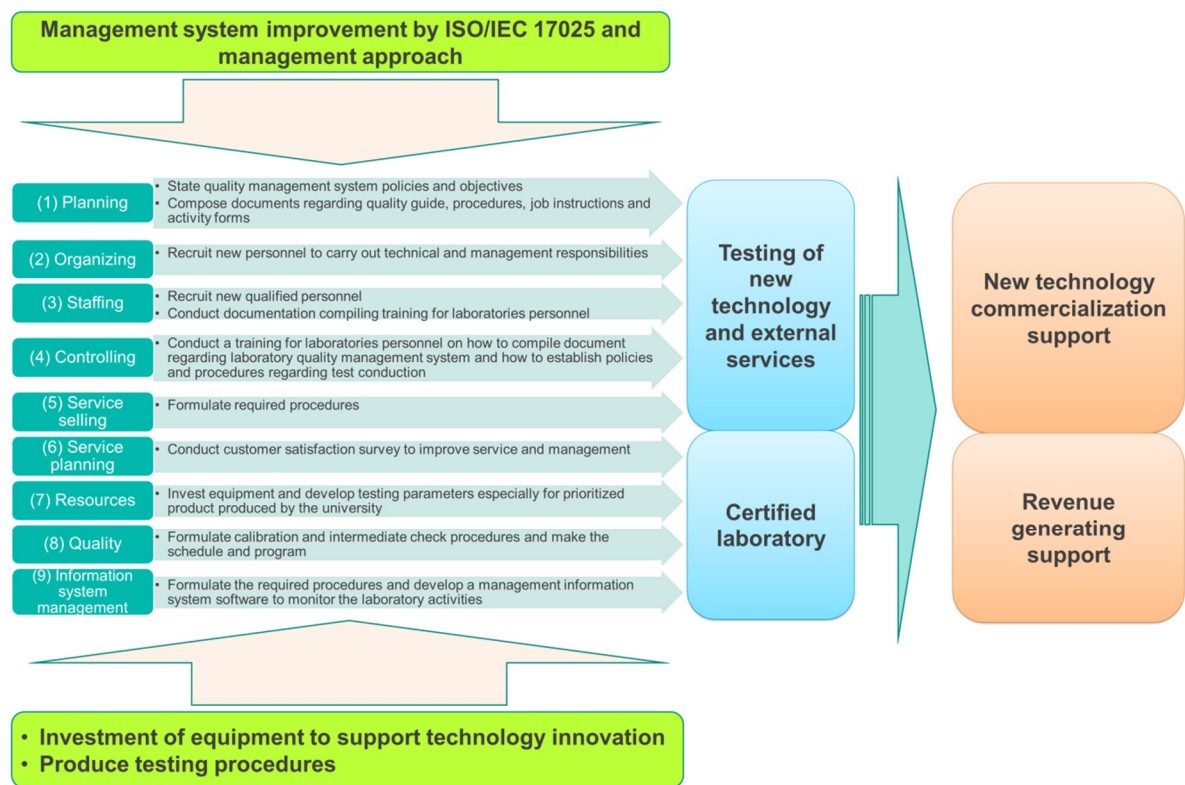

**Figure 6.** Summary of research implications.

The proposed framework can be used to monitor short-term and mid-term laboratory improvement to develop a certified laboratory. The measurement instrument can be used by the university body and policy-makers related to laboratory improvement to monitor the development from time to time; for instance, we can measure laboratory readiness at the beginning of the year and remeasure at the end of the year. Then, we can evaluate the gap between required conditions and existing conditions. If the gap could be minimized, we can decide to apply for accreditation. Based on the radar chart of laboratory readiness level, we can utilize the gap and propose the improvements depending on the attributes of 9 dimensions and 117 indicators.

From a management aspect, laboratories are ready to apply to be licensed testing laboratories if they have fulfilled ISO/IEC 17025 requirements. The investment of equipment that conforms to certain product testing requirements is needed to support WCU, product commercialization, and services business. For instance, the university has produced innovations of new technology for lithium batteries, electrical vehicles, and traceability. Therefore, equipment investment must be suitable for each technology product. In the case of the lithium battery product, the equipment has to comply with test parameters regulated in IEC 62660-2 and ISO 12405-1. Moreover, the university can be a certified agency after fulfilling the overall requirements. Therefore, the university would be able to assure the maximal contribution of technology commercialization and laboratory services. Future research can be done by formulating strategies to strengthen the nine dimensions in order to fulfill all of the required indicators.

## 5. Conclusions

This paper has developed a framework to assess university laboratory readiness to be developed as a testing laboratory to support the commercialization of new technology products resulting from the university innovation research. This research generated a measurement instrument to assess the laboratory readiness level, considering organizational audit factors in management concept and testing laboratory standard requirements. The instrument testing on 13 faculty laboratories

in the university indicated the laboratories' readiness level and organizational factors readiness. This research generated a radar chart of laboratory readiness level mapping and proposed improvements to strengthen the laboratory management system in order to accelerate towards accreditation application. Therefore, laboratories can operate as certified laboratories to provide standard services and generate certified products so the commercialization of the innovation product and revenue-generating role can be optimized.

**Author Contributions:** Conceptualization, W.S.; methodology, R.Z.; survey, E.F.A.; formal analysis, E.F.A.; writing—original draft preparation, E.F.A.; writing—review and editing, W.S. and E.F.A; supervision, W.S.; project administration, W.S., R.Z.

**Funding:** Institute for Research and Community Services, Universitas Sebelas Maret.

**Acknowledgments:** This research is supported by Institute for Research and Community Services, Universitas Sebelas Maret with Hibah Penelitian Mandatory' Research Program (Contract No. 543/UN27.21/PP/2018).

**Conflicts of Interest:** The authors declare no conflict of interest.

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
