# Peer review of "Model to Measure the Readiness of University Testing Laboratories to Fulfill ISO/IEC 17025 Requirements (A Case Study)"

_2199-8531, doi:10.3390/joitmc5010002_

Round 1
Reviewer 1 Report
The article is interesting in its research problem, and it has potential to provide useful practical implications to university research.
But, I could not find any relation with open innovation, the theme of the journal.
I hope the author(s) can improve the linkage with the article and the theme of the journal by reviewing and citing proper researches from the journal in the revision process.
Author Response
Point 1: The article is interesting in its research problem, and it has potential to provide useful practical implications to university research. But, I could not find any relation with open innovation, the theme of the journal. I hope the author(s) can improve the linkage with the article and the theme of the journal by reviewing and citing proper researches from the journal in the revision process 

Response 1: Description regarding to open innovation was added in the abstract, introduction, discussion and conclusion to declare linkage with the theme of journal, it can be found in the revised article in Line 5-7, Line 25-33, Line 260-287, Line 292-293, and line 299-301. We have reviewed and cited articles from the journal, and declared in line 26-27
Reviewer 2 Report
Research limitations have to be indicated.
List of literature should be extended. Suggested sources:
Zemguliene, J.; Valukonis, M. 2018. Structured literature review on business process performance analysis and evaluation, Entrepreneurship and Sustainability Issues 6(1): 226-252. https://doi.org/10.9770/jesi.2018.6.1(15)
Radwan, A. 2018. Science and innovation policies in North African Countries: Exploring challenges and opportunities, Entrepreneurship and Sustainability Issues 6(1): 268-282. https://doi.org/10.9770/jesi.2018.6.1(17)
Author Response
Point 1: Research limitations have to be indicated 

Response 1: We have declared the research limitations in the revised manuscript in Line 91-94
Point 2: List of literature should be extended 

Response 2: We have cited more literatures in the introduction of the revised manuscript in Line 26-27
Reviewer 3 Report
The topic researched is of great practical significance. I'd like to share several major concerns.
First, the single case study of a university (and laboratories within it) threatened the generalizability of your finding, which is further harmed by the unclear introduction to every lab (only grouped simply). It would be more contributive if we readers could know about the detail of the university, every lab, and their interactive relationships as a multilevel nested eco-system for industrial testing. The statistics presented in tables and figures seem less meaningful without knowing the details of every lab.
Second, it reads like that the 9 dimensions have been "determined" but not "emergent" through your data collection and analyses. These dimensions are relatively intuitive that the discussions of its special points that require us to find out from a published scientific work need to be articulated well.
Third, the paper fell short in providing theoretical and practical implications.
Author Response
Point 1: First, the single case study of a university (and laboratories within it) threatened the generalizability of your finding, which is further harmed by the unclear introduction to every lab (only grouped simply). It would be more contributive if we readers could know about the detail of the university, every lab, and their interactive relationships as a multilevel nested eco-system for industrial testing. The statistics presented in tables and figures seem less meaningful without knowing the details of every lab.

Response 1: The details of every lab were declared generally in Line 140. The interactive relationships of the labs for industrial testing were declared in discussion section of the revised manuscript in Line 260, Line 262-270 and Line 279-289.
Point 2: Second, it reads like that the 9 dimensions have been "determined" but not "emergent" through your data collection and analyses. These dimensions are relatively intuitive that the discussions of its special points that require us to find out from a published scientific work need to be articulated well.

Response 2: The 9 dimensions are the same as the sub criterions used in measurement instrument. We have given hints by adding “(SC1), (SC2)“ etc after mentioning the dimension in Line 224 of the revised manuscript
Point 3: Third, the paper fell short in providing theoretical and practical implications
Response 3: We have explained the implication of the study in discussion section in the revised manuscript in Line 231-289.
Round 2
Reviewer 1 Report
The paper is now acceptable for publication in JOItmC.
Author Response
Response 1: We have done English language and spell check
Reviewer 3 Report
I am happy with the revision the author(s) made. As an extra suggestion, if you could formalize your paper by grounding on the Ecosystem theory in Organization, that would be better for your theoretical value.
Author Response
Point 1: I am happy with the revision the author(s) made. As an extra suggestion, if you could formalize your paper by grounding on the Ecosystem theory in Organization, that would be better for your theoretical value
Response 1: We have added an introduction paragraph related to the ecosystem theory in organization (line 85-92). We also provide some discussions regarding this matter in the discussion section and this can be found in line 292-309.